# Improving the Efficiency and Safety of Sentinel Stink Bug Eggs Using X-rays

**DOI:** 10.3390/insects15100767

**Published:** 2024-10-04

**Authors:** Evelyne Hougardy, Ronald P. Haff, Brian N. Hogg

**Affiliations:** 1Invasive Species and Pollinator Health Research Unit, United States Department of Agriculture-Agricultural Research Services, Albany, CA 94710, USA; brian.hogg@usda.gov; 2Foodborne Toxin Detection and Prevention Research Unit, United States Department of Agriculture-Agricultural Research Services, Albany, CA 94710, USA; ron.haff@usda.gov

**Keywords:** biological control, egg parasitoids, pentatomids, field monitoring, irradiation, egg sterilization

## Abstract

**Simple Summary:**

The field monitoring of parasitic wasps attacking the eggs of stink bug pests is usually performed using sentinel eggs, i.e., stink bug eggs reared in a laboratory, placing them temporarily in the field. Sentinel eggs must be retrieved before the eggs hatch to avoid releasing the stink bug pest in the monitored area. In this study, X-ray irradiation was used to sterilize eggs of the bagrada bug, an invasive stinkbug pest of cole crops (cabbage, broccoli, cauliflower, kale, etc.) in the United States. Two parasitic wasps are known to attack bagrada bug eggs in California. Both wasps were able to parasitize irradiated eggs. Our results showed that X-ray irradiation is a suitable method to produce safe and reliable sentinel eggs to monitor the egg parasitism of the bagrada bug and possibly other species.

**Abstract:**

Sentinel eggs used to monitor field parasitism of stink bug pests (Hemiptera: Pentatomidae) can only be deployed for a few days to avoid releasing the pest in the monitored area. Using sterile eggs removes the risk of accidental pest introduction and extends deployment time. Freezing the eggs before deployment is one common method of sterilizing sentinel eggs. However, some egg parasitoid species have low or no parasitism on frozen eggs. In this study, X-ray irradiation was used to sterilize *Bagrada hilaris* sentinel eggs intended for monitoring parasitism by *Gryon aetherium* (Hymenoptera: Scelionidae), the most promising biological control candidate. In this case, freezing sentinel eggs is not recommended because *G. aetherium* has low levels of parasitism on frozen eggs. Doses as low as 10 Gy induced 100% sterility. Irradiated eggs successfully sustained the development of *G. aetherium* and *Ooencyrtus californicus* (Hymenoptera: Encyrtidae), another egg parasitoid attacking *B. hilaris*, and parasitism levels were comparable to that of fresh eggs up to seven days old. In addition, *G. aetherium* showed no preference for fresh non-irradiated eggs over seven-day-old irradiated eggs. Our results indicate that X-ray irradiation is a suitable alternative to produce safe and reliable sentinel eggs to monitor the egg parasitism of *B. hilaris* and possibly other species.

## 1. Introduction

The implementation of biological control programs requires the ability to monitor the presence and impact of biocontrol agents [1,2]. The biocontrol of stink bug pests (Hemiptera: Pentatomidae) often relies on egg parasitoids [3,4,5,6]. Egg parasitism in the field is monitored using sentinel eggs, either naturally occurring or laboratory-reared. Although wild sentinel eggs can lead to better estimates of the parasitism levels and species richness of the indigenous parasitoid community [7,8], laboratory-reared eggs are more commonly used because searching for egg masses in the field is time-consuming and not always practical. Laboratory-reared sentinel eggs are deployed in the field for a limited period, long enough to maximize the opportunity to be discovered by naturally occurring parasitoids, but not long enough that the parasitoids complete their development and emerge before the eggs are retrieved. Although more practical, laboratory-reared sentinel eggs come with the risk of releasing the pest in the monitored area when the unparasitized eggs hatch, thus limiting their deployment to 2 to 6 days [9,10,11,12]. In addition, egg parasitoids attacking stink bug eggs can usually only successfully parasitize eggs in their early stages of development. Being able to extend the duration of deployment could significantly improve the detection of parasitoids and even hyperparasitoids of stink bug pests [13].

Frozen eggs are commonly used for sentinel eggs [10,14,15]. Freezing the eggs kills the developing embryo while still providing a suitable food source for parasitoid development [16,17]. However, some parasitoid species have higher parasitism rates on frozen eggs compared to living eggs, a bias caused by the fact that freezing the eggs suppresses the host’s immune response to parasitism [18]. In contrast, other species have lower to no parasitism on frozen eggs due to rejection by the ovipositing female [19], or, in some cases, the lack of viability and development of the host needed for successful parasitism [20]. Furthermore, frozen eggs tend to start to decompose and desiccate a few days after thawing and are suitable for parasitoid oviposition for only a few days [21].

Ionizing radiation is the most commonly used method of inducing sterility in insects, a technique at the core of the sterile insect technique (SIT). This pest management practice consists of mass rearing, sterilizing, and releasing males of a pest species to compete with males in the wild population for mating opportunities [22]. Ionizing radiation can also be used to sterilize eggs directly without altering their suitability for egg parasitoid development [23]. Gamma-based irradiators have been traditionally used for insect sterilization, but are becoming more difficult to procure and maintain because of economic, regulatory, and safety concerns [24]. Recent studies show that X-rays are as effective as gamma rays in inducing sterility in insects [25,26]. There is increasing pressure to phase out gamma irradiators in favor of X-ray generating equipment, and the 2019 Nuclear Defense Authorization Act (Section 3141) calls for replacing all cesium-137 blood irradiators with X-ray equipment by 2027 [27].

This study was developed to improve the sentinel eggs deployed to monitor the egg parasitism of *Bagrada hilaris* (Burmeister) (Hemiptera: Pentatomidae), a worldwide pest of cole crops (broccoli, kale, cabbage, cauliflower, and all cultivated varieties of *Brassica oleracea* L.) found in the USA, Mexico, Chile, and Europe [28,29,30,31,32]. Nymphs and adults feed on various cultivated and wild brassicaceous species [33]. Conventional control strategies are usually ineffective because the natural habitats surrounding crops often act as a refuge for the pest. In addition, using conventional insecticides comes with economic and ecological costs, beyond the risk of insects developing resistance [34]. Therefore, biological control is considered a more sustainable control option for the bagrada bug, creating a demand for safe and reliable sentinel eggs to monitor the presence and impact of possible biocontrol agents, both in the region of origin of the pest or in invaded areas. Collecting naturally parasitized *B. hilaris* eggs is challenging because females lay most of their eggs under the soil surface [35], a unique strategy for a pentatomid species, most likely developed to escape egg parasitism. However, one egg parasitoid, *Gryon aetherium* Talamas (Hymenoptera: Scelionidae) has developed the ability to search for and parasitize eggs under the soil surface, and is currently the most promising biological control candidate for this pest [36]. Although originally identified in Pakistan, adventive populations of this parasitoid are now present in California [37]. Unfortunately, using frozen eggs to monitor *B. hilaris* field parasitism is not recommended because *G. aetherium* has low levels of parasitism on frozen eggs [19,38].

In this study, we investigated the feasibility of using X-ray irradiation to sterilize *B. hilaris* sentinel eggs. We first determined the doses needed to induce 100% sterility and if radiosensitivity decreased with egg age as observed in other insect species [23,39]. Next, we checked the suitability of irradiated eggs for the oviposition and development of two parasitoids: *G. aetherium* and *Ooencyrtus californicus* Girault (Hymenoptera: Encyrtidae), another egg parasitoid attacking *B. hilaris* found in California [37]. Finally, the preference of *G. aetherium* for irradiated eggs vs. fresh untreated eggs was investigated using choice tests. While this study was developed to enhance the field monitoring of *B. hilaris* parasitoids, the results could apply to other species for which frozen sentinel eggs are not a good option.

## 2. Materials and Methods

### 2.1. Insect Colonies

A laboratory colony of *B. hilaris* was established from field-collected individuals (Solano and Monterey counties, CA in 2015 and 2018) and maintained in bugdorm cages (61 × 61 × 61 cm, BioQuip Products Inc., Rancho Dominguez, CA, USA) at 29.0 ± 2.0 °C, 41 ± 5% RH, and a 16L:8D photoperiod. The colony was fed with store-bought organic broccoli and kale. Sand-filled (Quikrete Brown Play Sand, American Canyon, CA, USA) Petri dishes (diameter: 47 mm, height: 5 mm) sheltered below a laser-cut Plexiglas shade structure (14 × 7 × 2.5 cm) were provided as oviposition sites. Eggs were produced in separate oviposition boxes consisting of custom laser-cut ventilated Plexiglas boxes (28 × 16.5 × 14 cm), either oviposited in sand-filled Petri dishes (sand-covered) or on strips of cheesecloth (sand-free). Eggs were collected by sieving the sand using a No. 35 sieve (Humboldt Manufacturing Co., Elgin, IL, USA; mesh size: 500 μm) or by manually detaching the eggs from the cheesecloth using featherweight forceps. For the experiments below, we refer to fresh eggs for eggs that were laid within 24 h of their collection, “1-day-old” for eggs that were laid 24 to 48 h prior to their collection, and so on.

Egg parasitoid colonies were established with adults emerging from field-parasitized *B. hilaris* eggs collected near Davis, California, in 2020. The adults were kept in glass vials (diameter: 25 mm, height: 95 mm) at 20–24 °C, 40–60% RH, and a 16L:8D photoperiod, and fed organic raw honey spread on the inside of the stopper. About 2–4 times per week, the adults were exposed to one egg card consisting of 30–50 fresh (<24 h old) *B. hilaris* eggs glued (Elmer’s School Glue, Columbus, OH, USA) to a rectangular piece of card (20 × 60 mm). These egg cards were removed 1–3 days later and incubated in the same conditions until adults emerged 21 to 30 days later. Experimental parasitoid females were assumed to have mated, because they emerged in the presence of males and were housed with males within one day of their emergence until used for experiments. Organic raw honey was spread on the inside of the stoppers in the holding and experimental vials as a food source for the wasps.

### 2.2. Irradiation Process

The eggs were irradiated in a custom X-ray irradiator adapted from previously described versions [40,41] consisting of a shielded cabinet enclosing a pair of 1000 Watt fan beam X-ray tubes (MXR-100HP/20 FB, Comet Technologies USA Inc., Stamford, CT, USA) mounted side by side so that the emitted fan-shaped X-ray beams fell in a straight line on the surface of an adjacent rotating drum, parallel to the axis of rotation (Figure 1). X-ray tubes were energized with 1000-Watt (100 kV, 10 mA) power supplies (XPgN100, Matsusada Precision Inc., Kusatsu Shiga, Japan) and cooled with a portable chiller (M1-1.5A, Advantage, Greenwood, IN, USA). The tubes were strategically spaced so that the X-ray dose along the line was uniform, and thus, given many drum rotations, the dose at all points on the surface approached uniformity. The eggs were enclosed in 1-inch Ziplock bags and attached to the rotating drum with Velcro strips. The absorbed dose was measured with a radiation measurement system (Accu-Dose MNL/2086, Radcal Corporation, Monrovia, CA, USA) using a high-dose-rate ion chamber (10X6-0.18, Radcal Corporation) attached to the surface of the drum, allowing for real-time dose determination. The drum was rotated at 25 revolutions per minute. The dose rate applied varied depending on the target dose but was generally between 5 and 6 Gy/min (grays per minute). The irradiator settings in terms of voltage (kV) and current (mA) are given for each experiment.

### 2.3. Effect of X-ray Irradiation on Egg Eclosion

Fresh (<24 h old) *B. hilaris* eggs were irradiated at 0 (control), 10, 20, 30, and 40 Gy (81 kV; 6 mA; dose rate: 5.0 Gy/min). There were 30 eggs per dose. Both sand-covered and sand-free eggs were tested to determine if sand particles adhering to the egg surface serve as shielding against irradiation. After irradiation, the eggs were transferred into glass vials and incubated at 23.5 ± 0.3 °C and 69 ± 9% RH. Freshly oviposited *B. hilaris* eggs are whitish/cream-colored, then slowly turn orange as they develop and become bright orange before eclosion. Freshly eclosed nymphs are bright orange (Figure 2). At 23 °C, nymphs started to emerge on day 7. In this experiment, the eggs were visually checked 12 days post irradiation and categorized as either (1) eclosed, (2) dead and uneclosed but showing some development (orange tint/color), or (3) dead and uneclosed with no embryo development (cream color).

### 2.4. Effect of Egg Age on Radiosensitivity

In a similar experiment, we irradiated 1 d (<24 h), 2 d (24–48 h), and 3 d (48–72 h) old *B. hilaris* eggs, as well as eggs refrigerated at 8 °C for 7 days, with a dose of 40 Gy (81 kV; 6 mA; dose rate: 5.0 Gy/min). Non-irradiated controls were also included for each treatment. After the irradiation, the eggs were transferred into glass vials and incubated in the same condition as above. The eggs were visually checked 9 days post irradiation and categorized as either (1) eclosed, (2) dead and uneclosed but showing some development (orange color/tint), or (3) dead and uneclosed with no embryo development (cream color).

### 2.5. Suitability of Irradiated Eggs for Parasitoid Development

Parasitism by *G. aetherium* was tested in no-choice tests on fresh (<24 h old), 7-, and 12-day-old irradiated eggs (40 Gy; 81 kV; 7 mA; dose rate: 5.8 Gy/min). The controls were fresh non-irradiated eggs. An additional control with 7-day-old eggs was also included (no control was available for 12-day-old eggs because *B. hilaris* egg eclosion usually starts 7–8 days post oviposition at 23 °C). Ten egg cards with ten sand-covered eggs each were prepared for each treatment (ten replicates) and exposed to a single 1- to 5-day-old *G. aetherium* wasp for 24 h. The numbers of emerged *B. hilaris* nymphs, emerged parasitoids and their sex, and dead unemerged eggs were recorded after 30 days.

Parasitism by *O. californicus* was tested similarly with 4- to 6-day-old females using a no-choice test with 7-day-old irradiated eggs and a control consisting of fresh (<24 h old) non-irradiated eggs. There were 16 replicates (16 egg cards). Because parasitism rates were lower with this species, exposure was extended from 24 to 48 h for the last 6 replicates.

### 2.6. Parasitoid Preference

The preference of *Gryon aetherium* for fresh non-irradiated vs. 7-day-old, irradiated eggs was tested in choice tests, in which single 2- to 4-day-old parasitoid females were presented with five fresh eggs and five 7-day-old irradiated eggs. The two types of eggs were glued on to cards in two rows of 5. After 24 h of exposure, the egg cards were cut longitudinally, and their halves, each containing 5 eggs of one type, were incubated in separate vials. There were 10 replicates. The numbers of emerged *B. hilaris* nymphs, emerged parasitoids and their sex, and dead unemerged eggs were recorded after 30 days.

### 2.7. Data Analysis

Parasitism rates were calculated as the sum of emerged parasitoids divided by the number of eggs. We assumed that no more than one adult emerged from each host egg for both species. Indeed, *G. aetherium* appears to avoid conspecific superparasitism [42]. And, while some *Ooencyrtus* spp. can sometimes be facultatively gregarious [43], when this happens, we should expect the size of the co-emerged adults to be considerably reduced. Since no obvious difference in size was observed among *O. californicus* adults in our experiments, we rejected the possibility of gregarious development. The sex ratio was expressed as the proportion of females and was calculated as the number of female parasitoids divided by the total number of parasitoids. Proportion parasitism and proportion females were compared between treatments using generalized linear models (GLMs) with binomial errors, using the GLM function in R version 4.2.3 [44]. For the experiment with *O. californicus* and the choice test with *G. aetherium*, the treatments were compared using mixed model GLMs (GLMMs), including either the trial date (for the experiment with *O. californicus*, to account for differences in exposure time between trial dates) or replicate (for the choice test with *G. aetherium*, to account for the non-independence of the data from replicates) as a random factor, using the glmer function in R. Tukey’s HSD tests were used for multiple comparisons. The dataset supporting the findings of this study can be found in Appendix A.

## 3. Results

### 3.1. Effect of X-ray Irradiation on Egg Eclosion

X-ray doses as low as 10 Gy induced 100% sterility (no eclosion) (Figure 2). A proportion of irradiated eggs turned a light orange tint, suggesting that some embryo development may have occurred. A higher proportion of these eggs showing signs of development were recorded at lower doses, and/or in sand-free eggs. Overall, the irradiated eggs started deteriorating at 8 days post irradiation, the chorion progressively separating from the shell and slowly shrinking over time. By day 14, approximately 75% of the eggs appeared desiccated and unlikely to sustain parasitoid development. Twelve days post-irradiation, most control eggs (0 Gy) had eclosed: 98% and 89% of sand-free and sand-covered eggs, respectively. The remaining uneclosed control eggs did not develop or contained dead unemerged nymphs (natural mortality).

### 3.2. Effect of Egg Age on Radiosensitivity

Egg hatching was prevented in all irradiated 1-, 2-, and 3-day-old treatments (Figure 3). The proportion of eggs showing some embryo development increased with egg age, while no development was observed in 7-day-old irradiated refrigerated eggs. Over 85% of nymphal development occurred in all non-irradiated control treatments.

### 3.3. Suitability of Irradiated Eggs for Parasitoid Development

There was no difference in *G. aetherium* parasitism rate (GLM, χ^2^ = 0.58; df = 1; *p* = 0.45) nor sex ratio (χ^2^ = 0.02; df = 1; *p* = 0.90) between fresh irradiated and non-irradiated eggs (Figure 4). The parasitism rate differed between fresh non-irradiated, 7-day-old non-irradiated, and 7-day-old irradiated eggs (χ^2^ = 195.75; df = 2, *p* < 0.001), and was significantly lower for 7-day-old non-irradiated compared to fresh non-irradiated or 7-day-old irradiated eggs (Tukey’s HSD, *p* < 0.05). There was also a significant difference in overall sex ratio (χ^2^ = 7.71; df = 2; *p* = 0.02), with fewer females produced on irradiated eggs, although there were no significant differences between treatments in pairwise comparisons (Tukey’s HSD, *p* > 0.05). A proportion of the 7-day-old non-irradiated eggs had already hatched on the day of the exposure (7%), and more hatched the next day (41%). The uneclosed eggs most likely contained fully developed nymphs ready to eclose. Surprisingly, about 15% of these eggs still provided a suitable resource for parasitoid development. Parasitism on 12-day-old irradiated eggs was lower compared to fresh eggs (χ^2^ = 13.27; df = 1; *p* < 0.001), but there was no difference in sex ratio (χ^2^ < 0.01; df = 1; *p* = 0.95). Parasitism by *O. californicus* females was greater on irradiated compared to control eggs (GLMM, χ^2^ = 33.21; df = 1; *p* < 0.001), while there was no significant difference in sex ratio (χ^2^ = 0.79; df = 1; *p* = 0.37) (Figure 5).

### 3.4. Parasitoid Preference

When *G. aetherium* females were given a choice between fresh untreated eggs and 7-day-old irradiated eggs, they did not show any preference: there was no difference in parasitism rates (GLMM, χ^2^ = 0.06; df = 1; *p* = 0.81). There was, however, a difference in sex ratio: more females were produced on irradiated compared to control eggs (χ^2^ = 4.48; df = 1; *p* = 0.03) (Figure 6).

## 4. Discussion

Our results showed that X-ray doses as low as 10 Gy prevent *B. hilaris* eggs from hatching. When no embryo development is preferred, 40 Gy and higher doses are preferable. Our results also demonstrated that the sand particles adhering to the egg surface provide some protection from the radiation, although not enough to compromise the efficiency of the sterilization at the doses we tested. This is promising, since sieved eggs are less time-consuming to collect. In a similar study undertaken with the brown marmorated stink bug (BMSB) *Halyomorpha halys* (Stål) (Hemiptera: Pentatomidae) using gamma rays, doses as low as 16 Gy induced 100% egg sterility [23]. A large body of information on the radiosensitivity of insect eggs is available for lepidopteran pests. These studies usually also use gamma radiation and show that much higher doses are needed to induce 100% sterility compared to X-rays. For instance, doses of 300 to 350 Gy were required to inhibit the development of *Ephestia cautella* (Walker) eggs [45], while *Plodia interpunctella* (Hübner) (Lepidoptera: Pyralidae) eggs require 450 Gy [46]. This apparent inconsistency in the sterilization dose required for eggs of different insect species has many possible explanations, including non-equivalency in the biological effects of X-rays and gamma rays, different dosimetry methods, and/or differences in the density and structure of the chorion of lepidopteran vs. hemipteran eggs [47]. For this study, all sterilization doses reported refer to the dose as measured with the reader and probe described under the X-ray irradiation parameters given for each experiment. Different dose rates, irradiation methods, and dosimetry tools could lead to different results in terms of the sterilization doses required and the subsequent viability of the irradiated eggs.

Our results did not indicate a correlation between radiosensitivity and egg age. Some embryo development was observed in older eggs, but this is most likely due to the normal embryogenesis taking place before the irradiation. Indeed, no such development was observed in eggs kept at 8 °C for 7 days, because the cooler temperature slowed down or prevented embryogenesis. In contrast, 2-day-old BMSB eggs were more radioresistant than fresh eggs, requiring doses of 40 Gy and above to induce 100% sterility [23]. A similar decrease in egg radiosensitivity with age was also observed with lepidopteran eggs, as demonstrated for the Mediterranean flour moth, *Ephestia kuehniella* (Lepidoptera: Pyralidae): doses of 75, 125 and 350 Gy were needed to induce 100% sterility in 1-, 2-, and 3-day-old eggs, respectively [39]. Developmental changes may have provided partial protection from irradiation for BMSB and lepidopteran eggs, but not for bagrada bug eggs.

X-ray irradiated eggs were suitable for *G. aetherium* and *O. californicus* oviposition and development. Irradiated eggs performed similarly to fresh non-irradiated eggs for *G. aetherium* parasitism for at least 7 days. *Bagrada hilaris* nymphs usually complete their embryonic development by day 7 (at 23 °C) and start to emerge. These eggs are therefore no longer suitable for parasitoid development. The reported 15% parasitism on untreated 7-day-old eggs was probably related to the parasitism of unfertilized eggs or eggs with dead undeveloped embryos. In any case, by testing 7-day-old irradiated eggs, we showed that irradiating the eggs extends their period of vulnerability to parasitism. Similarly, irradiated BMSB eggs were suitable for the development of *Trissolcus japonicus* (Ashmead) (Hymenoptera: Scelionidae), the most promising candidate for the classical biological control of BMSB, with parasitism rates reaching 84–91% during the first 7 days before declining at day 10 (no control with fresh eggs was provided) [23]. *Ooencyrtus californicus* performed even better on irradiated eggs compared to untreated eggs, possibly due to the lack of immune response due to the absence of a developing embryo, as has been observed for other generalist egg parasitoids on frozen eggs [14,18]. This suitability of irradiated eggs to parasitoid development is not always observed. For instance, *Gryon gnidus* (Nixon) would not parasitize the gamma-irradiated eggs (10 Gy) of its host *Acanthomia tomentosicollis* (Stål) (Hemiptera: Coreidae) [20].

In our study, we used sex ratio as a proxy for host quality. Indeed, hymenopteran parasitoids with haplodiploid reproductive systems tend to allocate male eggs to hosts that they assess to be of lower quality [48]. In most of our experiments, the sex ratio was not significantly different in irradiated eggs vs. fresh eggs. This suggests that the host quality of irradiated eggs, or at least of the irradiated eggs accepted for oviposition, remained comparable to that of fresh eggs. The same observation was made with *T. japonicus* when parasitizing irradiated BMSB eggs: the sex ratio hovered around 82–84% up to day 7, which is comparable to the sex ratio in non-irradiated fresh BMSB eggs (usually > 83%) [49], before starting to decline at day 10 [23].

## 5. Conclusions

Our study showed the efficiency of X-rays in producing safe and functional sentinel eggs of *B. hilaris*. These irradiated eggs could be deployed for at least 7 days with an efficiency comparable to that of fresh eggs. After 7 days, they could still be employed to evaluate parasitism with no risk of accidental pest release. Future research will focus on field testing irradiated sentinel eggs and comparing their efficiency in capturing resident parasitoid species with that of non-irradiated eggs.

## Figures and Tables

**Figure 1 insects-15-00767-f001:**
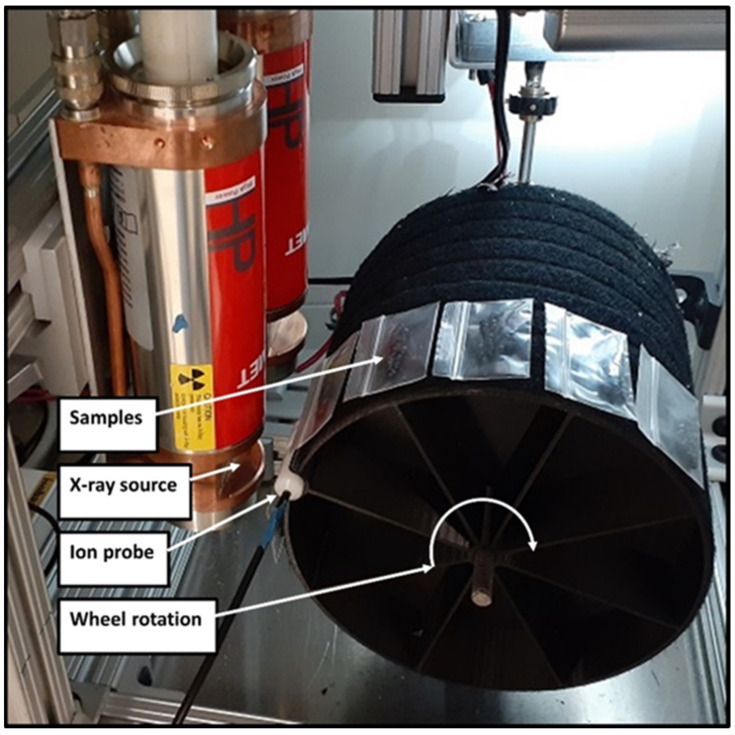
Interior of the X-ray irradiator showing samples in Ziploc bags rotating adjacent to the radiation source. Real-time dose measurements from the attached ion probe were read from an external display, and X-rays were discontinued when the desired dose was reached.

**Figure 2 insects-15-00767-f002:**
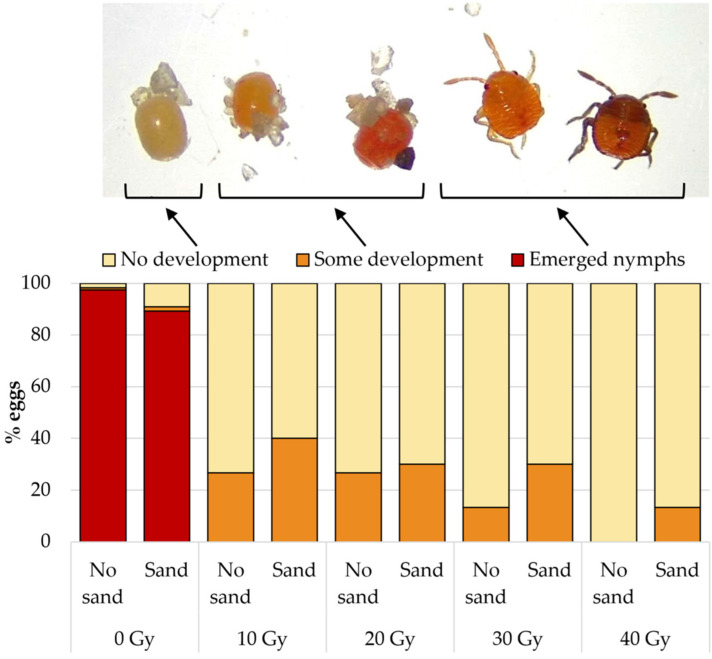
Influence of X-ray irradiation (0–40 Gy) on the percentage of sand-free (“no sand”) and sand-covered (“sand”) *B. hilaris* eggs showing no or some embryo development or eclosed 12 days post irradiation.

**Figure 3 insects-15-00767-f003:**
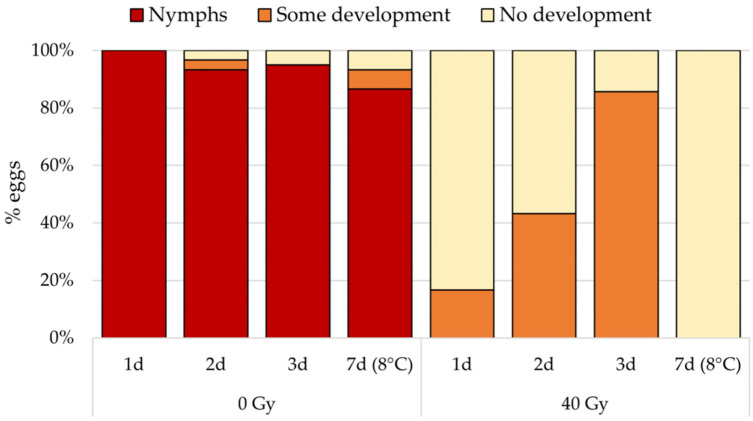
Percentage of 1, 2, 3, and 7 d old (refrigerated) eggs showing no or some embryo development, or eclosed 9 days post irradiation (40 Gy).

**Figure 4 insects-15-00767-f004:**
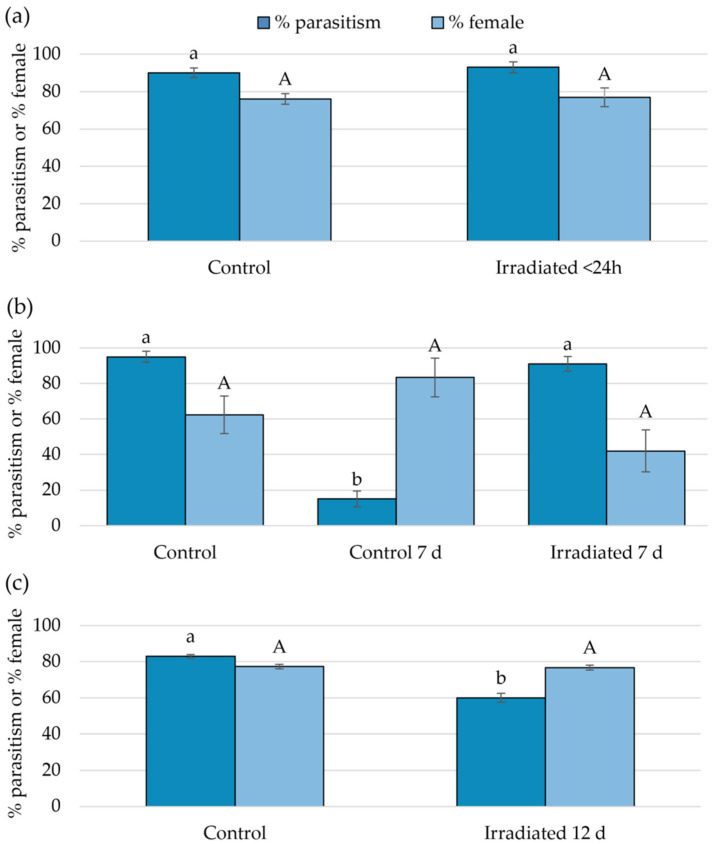
Percentage of parasitism and sex ratio (percentage female) by *G. aetherium* on (**a**) fresh (<24 h old), (**b**) 7-, and (**c**) 12-day-old irradiated (40 Gy) *B. hilaris* eggs. Controls were fresh *B. hilaris* eggs, with the addition of one control consisting of 7-day-old non-irradiated eggs when testing 7-day-old irradiated eggs. Bars with different letters are significantly different (GLM or Tukey’s HSD, *p* < 0.05).

**Figure 5 insects-15-00767-f005:**
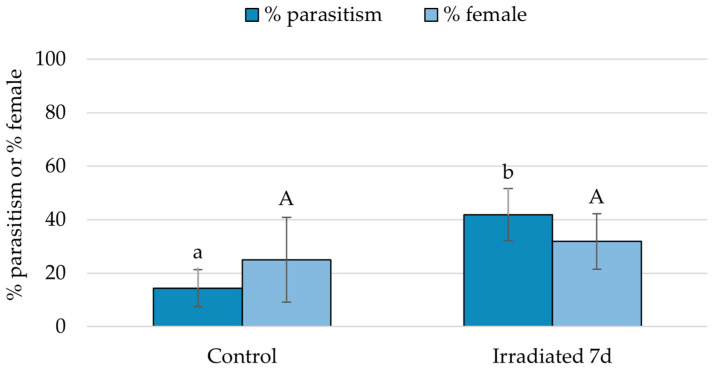
Percentage of parasitism and sex ratio (percentage female) by *O. californicus* on 7-day-old irradiated (40 Gy) *B. hilaris* eggs. Controls were fresh *B. hilaris* eggs. Bars with different letters are significantly different (GLMM, *p* < 0.05).

**Figure 6 insects-15-00767-f006:**
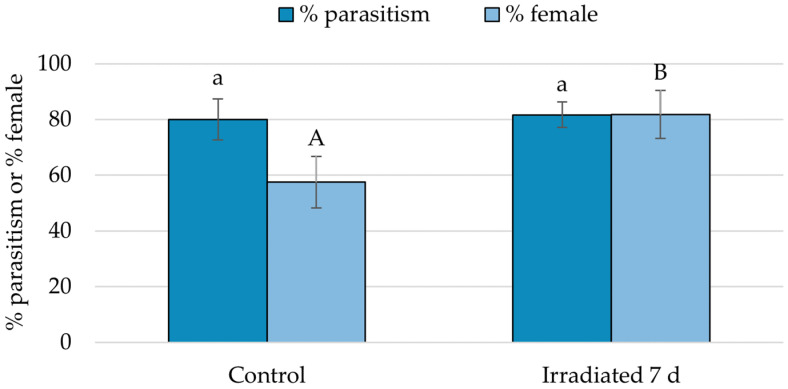
Percentage parasitism and female production when *G. aetherium* was given a choice between 7-day-old irradiated eggs (40 Gy) and fresh (<24 h old) *B. hilaris* eggs. Bars with different letters are significantly different (GLMM, *p* < 0.05).

## Data Availability

The data are contained within the article or the Appendix A.

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
