# Peer review of "Improving the Efficiency and Safety of Sentinel Stink Bug Eggs Using X-rays"

_insects, 2024, doi:10.3390/insects15100767_

Round 1

Reviewer 1 Report

Comments and Suggestions for Authors

The article has interesting and well-presented content. The authors included images and figures that allowed a good understanding of the work. I think it would be interesting if the authors justified the age of the eggs chosen.

Reviewer 2 Report

Comments and Suggestions for Authors

This paper details utilizing x-ray radiation to sterilize eggs of B. hilaris for use as sentinel eggs in the field. I thought this paper was well written and organized in a straight-forward and intuitive manner. I only had minor editorial comments, which were made in the body of the paper using the comments tool. I feel this work will be valuable to those working with biological control of stink bugs in agricultural and other systems.

Reviewer 3 Report

Comments and Suggestions for Authors

Hougardy et al. report the results of a laboratory study on the effect of x-ray irradiation of the viability of stick bug eggs and their acceptability as hosts for two egg parasitoids. The study appears to have been performed correctly and I have only minor suggestions for improvements.
The writing style can be improved, both in formality and clarity.
I have written suggestions and numbered points on a scanned copy of the manuscript.
NUMBERED POINTS (see scanned file)
1.    Do not use keywords that are already in the title.
2.    Please replace lab with “laboratory” throughout the manuscript.
3.    Point out that the SIT is based on the release of sterile MALES that compete with wild (fertile) males.
4.    Did emerging wasps have access to food (honey?) or water?
5.    The usual abbreviation for milliamps is “mA” (not ma). Also the x-ray energy is stated to be given in keV (line 143) but is subsequently given in kV in the subsequent sections (which I suspect is a machine set voltage).
6.    Again, did parasitoids have access to food or water during the test oviposition period?
7.    The paragraph on O. californicus mentions 16 replicates. But how many replicates were performed for G. aetherium (same number? Unclear).
8.    The overall sex ratio was significantly affected in favor of females in irradiated or non-irradiated eggs?
9.    You need to cite Fig 5 here, I think.
10.    The argument in this text cannot be understood without mention of the sex ratio in non-irradiated BMSB eggs.
11.    There are some references with missing information (see the ones that I spotted).
Also, thank you for supplying the raw data. This should be mentioned in the Data Availability Statement (line 326). Also, in the Excel file, could you mention the FIGURE (graph) that each Excel sheet refers to, just for clarity (a brief title at the top of each sheet would do).

Comments on the Quality of English Language

Some editing required (see scanned ms)

Reviewer 4 Report

Comments and Suggestions for Authors

In this work the authors investigate the effects of X-ray application in both IN viability of eggs and suitability for egg parasitoids.

In general, the work is rationally planed and executed. Methodology, results and discussion are clear and adequates.

Some minor comments will be find in the attached pdf file.

Sincerely

Raúl Laumann

Comments on the Quality of English Language

As english is not my mother lenguage I can not evaluate and comment this topic.
